# On the Interpretability of Regularisation for Neural Networks Through Model Gradient Similarity

**Vincent Szolnoky**
Department of Mathematical Sciences
Chalmers University of Technology
Göteborg, Chalmers Tvärgata 3, 41296, Sweden
szolnoky@chalmers.se

**Viktor Andersson**
Department of Electrical Engineering
Chalmers University of Technology
Göteborg, Chalmersplatsen 4, 41296, Sweden
vikta@chalmers.se

**Balázs Kulcsár**
Department of Electrical Engineering
Chalmers University of Technology
Göteborg, Chalmersplatsen 4, 41296, Sweden
kulscar@chalmers.se

**Rebecka Jörnsten**
Department of Mathematical Sciences
Chalmers University of Technology
Göteborg, Chalmers Tvärgata 3, 41296, Sweden
jornsten@chalmers.se

## Abstract

Most complex machine learning and modelling techniques are prone to overfitting and may subsequently generalise poorly to future data. Artificial neural networks are no different in this regard and, despite having a level of implicit regularisation when trained with gradient descent, often require the aid of explicit regularisers. We introduce a new framework, *Model Gradient Similarity* (MGS), that (1) serves as a metric of regularisation, which can be used to monitor neural network training, (2) adds insight into how explicit regularisers, while derived from widely different principles, operate via the same mechanism underneath by increasing MGS, and (3) provides the basis for a new regularisation scheme which exhibits excellent performance, especially in challenging settings such as high levels of label noise or limited sample sizes.

## 1 Introduction

Since the inception of modern neural network architecture and training, many efforts have been made to understand their generalisation properties. When neural networks are trained with Gradient Descent (GD) this induces implicit regularisation in the network, causing it to attain surprising generalisation performance with no extra intervention. Despite this, neural networks will in many instances overfit in the presence of noise and have been shown to have the capacity to completely memorise data [Zhang et al., 2016]. Therefore, alongside the desire to understand how neural networks are implicitly regularised, explicit regularisation has been an ongoing pursuit to improve generalisation even further.

A broad range of explicit regularisers now exist that attack the problem from many different angles. Below, we provide a short overview of some of the most commonly used methods. *Weight decay*

36th Conference on Neural Information Processing Systems (NeurIPS 2022).

[Krogh and Hertz, 1991] places a L2-penalty on the parameters to encourage a minimum-norm solution. *Normalisation schemes*, that normalise the data as it flows through the network, have been shown to act as explicit regularisers [Ioffe and Szegedy, 2015]. *Double back-propagation* incorporates the gradients with respect to the loss itself as part of the loss, hence the name, as the gradient of the gradient will be evaluated in the complete back-propagation step [Drucker and Le Cun, 1991]. Although these methods were initially used as a part of a energy minimisation principle, they are now believed to help finding "flat minima", which have been shown to produce solutions that generalise better [Zhao et al., 2022]. *Dropout* turns off connections between neurons during training and thus forces the network not to rely on any one given connection [Srivastava et al., 2014]. Another way to view the learning problem is to ignore the network itself and instead focus on the optimisation scheme. Here, a multitude of different learning rate and optimisers exist that offer explicit regularisation. Some standout examples include *Cyclical Learning Rate* [Li and Yang, 2020] and the *Adam* optimisation algorithm [Kingma and Ba, 2015]. More recently, especially in works related to the Neural Tangent Kernel (NTK) [Jacot et al., 2018], focus has been put onto functional regularisation. The core idea is to view the neural network architecture as being an opaque function approximator and instead regularise it as it is viewed from function space rather than the parameter space. This is usually achieved via a form of *kernel ridge regression* or an approximation thereof [Bietti et al., 2019, Hoffman et al., 2019].

**Our contribution**. In this article, a new framework called *Model Gradient Similarity* (MGS) is introduced. MGS builds on the idea that regularisation and, to a large extent, generalisation, is tightly connected to how the model evolves during training. We show that MGS can be used to analyse neural network training, monitoring how model gradients evolve together. It is revealed that explicit regularisers, despite their fundamentally different construction, operate via the same underlying mechanisms in terms of increasing model gradient similarity. This insight lays the foundation for a new class of regularisers aimed at directly boosting model gradient similarity. We derive two simple MGS regularisation techniques from first principles and compare performance to a wide range of regularisation methods. In a majority of the cases, the MGS regularisation achieves top performance and exhibits robustness qualities. This suggests that the MGS framework opens the door for the further development of both novel regularisers and the improvement of existing ones.

## 2 Model Gradient Similarity

Let $x \in X$ denote an input observation in a data batch and $y \in Y$ the target. We will train the model $f_\theta$, parameterised by $\theta$, using loss function $\mathcal{L}$ via gradient descent (GD). Thus, the standard single-observation batch update rule using GD at time $i$ is $\theta_{i+1} = \theta_i - \eta \nabla_\theta \mathcal{L}(f(x), y)$, where $\eta$ is the learning rate. When training a neural network, $f_\theta$ shall represent the raw output from the network before any additional normalisation is performed (e.g. softmax as in the case of classification).

The loss gradients with respect to the parameters $\nabla_\theta \mathcal{L}$ (**loss-parameter gradient**) are thus the main contributing factor to the update of the model. In the general case for common losses, such as mean-squared error or cross-entropy, we apply the chain rule to the loss-parameter gradient: $\nabla_\theta \mathcal{L} = \nabla_\theta f_\theta \cdot \nabla_f \mathcal{L}$. We denote the two components as: the loss gradients with respect to the model output $\nabla_f \mathcal{L}$ (**loss-model gradients**) and the model gradients with respect to the model parameters $\nabla_\theta f_\theta$ (**model-parameter gradients**).

Studying the influence that certain input data has on other data points during training is a common technique from robust statistics. The use of influence functions to study model behaviour has been explored for neural networks to, for example, help identify important training samples and detect mislabelled data [Koh and Liang, 2017]. How learning one data point influences another has also been used in algorithms for clustering data [He and Su, 2020]. Recently this idea has been broadened under the general concept of gradient similarity/coherence and has been suggested as an explanation for a neural network's ability to generalise [Faghri et al., 2020, Chatterjee and Zielinski, 2022]. It is here, however, mainly considered from the perspective of the loss-parameter gradient $\nabla_\theta \mathcal{L}$ and not the model-parameter gradients $\nabla_\theta f_\theta$. Contrary to the loss-parameter gradients, which diminish as any minima is approached, the model-parameter gradients display different behaviour and describe the evolution of the function realised by the model. To help understand why they can be useful in understanding neural network generalisation and regularisation thereof, we will borrow some inspiration about gradient similarity from Charpiat et al. [2019] and use it to define the core ideas behind MGS.

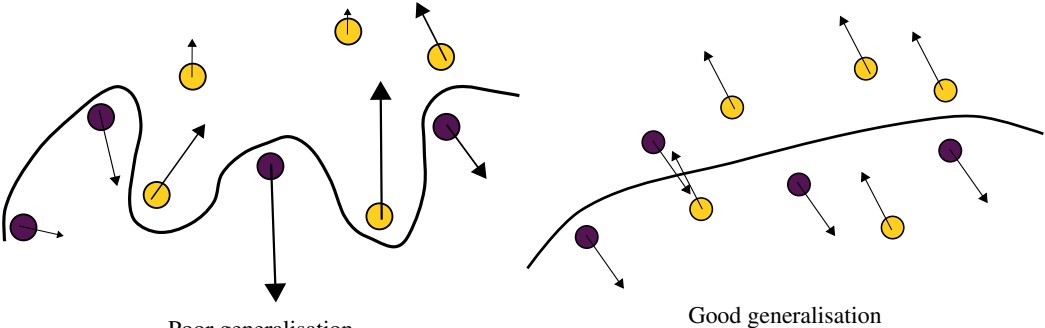

Poor generalisation

Good generalisation

Figure 1: **An illustration on the connection between MGS and generalisation.** The arrows represent the **model gradients**, $\nabla f_\theta$, for each data point. **Left:** A model that exhibits low similarity between its gradients. The model is free to adapt the gradients for each point and therefore learns them individually. The most likely outcome then is that it will overfit the data and generalise poorly. **Right:** A model that is required to maintain a level of similarity between its gradients. To lower the overall loss, while maintaining model gradient similarity, the model is forced to learn from the data in groups defined by similar gradients. Thus, the model will have to learn separations in the data based on such groupings instead of each point individually.

## 2.1 Defining MGS and its Relation to Generalisation

When judging the similarity of two points $x$ and $x'$ from the model's point of view, one may be inclined to simply compare the output $f_\theta(x)$ to the other $f_\theta(x')$. Unfortunately, due to the non-linear nature of neural networks, the same output might be produced for a number of different reasons and it is therefore difficult to draw conclusions about similarity directly from the output.

An alternative notion of similarity between $x$ and $x'$ can be defined by how much changing the model output for $x$ (by training on this sample) changes the output for $x'$. If the points are dissimilar from the model's standpoint, then changing $f_\theta(x)$ should have little influence over $f_\theta(x')$ and vice-versa. That is, after one step using GD for a single input sample $x$ the parameters are changed by:

$$\delta\theta = -\eta\nabla_\theta f \nabla_f \mathcal{L}\left(f(x), y\right).$$

This induces the following change in the output of the model for that specific $x$:

$$\begin{aligned}
f_{\theta+\delta\theta}(x) &= f_\theta(x) + \nabla_\theta f(x) \cdot \delta\theta + O(||\delta\theta||^2) \\
&\approx f_\theta(x) - \eta\left(\nabla_\theta f(x) \cdot \nabla_\theta f(x)\right)\nabla_f \mathcal{L}\left(f(x), y\right).
\end{aligned} \quad (1)$$

On the other hand, this update will also yield a change in the model for another $x'$ given by:

$$\begin{aligned}
f_{\theta+\delta\theta}(x') &= f_\theta(x') + \nabla_\theta f(x') \cdot \delta\theta + O(||\delta\theta||^2) \\
&\approx f_\theta(x') - \eta\left(\nabla_\theta f(x') \cdot \nabla_\theta f(x)\right)\nabla_f \mathcal{L}\left(f(x), y\right).
\end{aligned} \quad (2)$$

Therefore, the kernel $k_\theta(x, x') = \nabla_\theta f(x) \cdot \nabla_\theta f(x')$ (derived from the model-parameter gradients) is crucial to understanding how the model evolves after one step of GD. It determines how much of the loss-model gradient $\nabla_f \mathcal{L}\left(f(x), y\right)$ is used to update the model, up to a scaling by the learning rate. Another way to look at it is that $k_\theta(x, x')$ describes the influence learning $x$ has over $x'$. If $k_\theta(x, x')$ is large, which occurs when the model gradients are similar, then an update for $f(x)$ will also move $f(x')$ in the same direction. Likewise, if $k(x, x')$ is small, meaning the model gradients are dissimilar, then the update for $f(x)$ will have little affect on $f(x')$. The cartoon example in figure 1 illustrates how this property can be useful to understand how a model generalises better if it exhibits higher MGS.

## 2.2 The contribution of MGS to Gradient Descent

If equations 1 and 2 above are combined, the complete update for a single step of gradient descent with training data $X$, is:

$$\begin{aligned}
\mathbf{f}_{\theta+\delta\theta}(X) &= \mathbf{f}_\theta(X) + \nabla_\theta \mathbf{f}(X) \cdot \delta\theta + O(||\delta\theta||^2) \\
&\approx \mathbf{f}_\theta(X) - \eta K_\theta(X) \cdot \nabla_f \mathcal{L}\left(\mathbf{f}(X)\right) \\
&= \begin{bmatrix} f_\theta(x_1) \\ \vdots \\ f_\theta(x_n) \end{bmatrix} - \eta \begin{bmatrix} k(x_1, x_1) & \cdots & k(x_1, x_n) \\ \vdots & \ddots & \vdots \\ k(x_n, x_1) & \cdots & k(x_n, x_n) \end{bmatrix} \cdot \begin{bmatrix} \nabla_f \mathcal{L}(f(x_1), y_1) \\ \vdots \\ \nabla_f \mathcal{L}(f(x_n), y_n) \end{bmatrix}.
\end{aligned}$$

Here $K_\theta(X) = k(X, X)$ is the kernel matrix for data $X$. The update for each model output $f_\theta(x_k)$ can thus be seen as a weighted average of the loss-model gradients $\nabla_f \mathcal{L}\left(f(x_j), y_j\right)$:

$$f_{\theta+\delta\theta}(x_k) = f_\theta(x_k) - \eta \sum_{j=1}^n k(x_k, x_j) \cdot \nabla_f \mathcal{L}(f(x_j), y_j), \ x_k \in X. \tag{3}$$

From equation 3, it follows that the kernel ultimately controls the weighting of the loss-model gradients: the greater the similarity exhibited between the model gradients, the greater the averaging effect will be. This partially explains how GD induces implicit regularisation as, unless $K_\theta(X)$ is dominated by its diagonal, a model update for observation $x_k$ will also utilise the loss gradients of other observations.

Furthermore, it has been observed from the Neural Tangent Theory perspective that $K_\theta(X)$ will often align with the ideal kernel $YY^T$ in classification problems, which perfectly discriminates targets. This phenomenon has already been investigated as one of the reasons behind implicit generalisation for neural networks when trained with SGD [Kopitkov and Indelman, 2020, Baratin et al., 2021]. Here, we also see that the alignment will naturally engage the averaging effect, grouping gradients according to their target classes. It is still unclear what the cause of the alignment is. However, MGS provides an explanation as to why such an alignment is useful. More detail on the connection to Neural Tangent Theory is presented in the supplementary material.

To summarise, apart from capturing to what extent the model gradients are aligned with each other, MGS also determines how loss-model gradients for observations that have high MGS with each other are averaged in the GD step.

## 2.3 Metrics to Quantify MGS

Since the kernel $K_\theta(X)$ captures coordinated learning between similar observations (equation 3), it is desirable to identify metrics which can summarise the overall model gradient similarity in a batch of data.

To find relevant metrics of $K_\theta$ we note that the spectrum of $K_\theta(X) = (\nabla_\theta f(X))^T \nabla_\theta f(X)$ is the same as that of $\tilde{C} = \nabla_\theta f(X)(\nabla_\theta f(X))^T$; the non-centered version of the sample covariance matrix $C = (\nabla_\theta f(X) - \overline{\nabla_\theta f(X)})(\nabla_\theta f(X) - \overline{\nabla_\theta f(X)})^T$. The spectrum of $C$ and $\tilde{C}$ are interlaced as their corresponding eigenvalues $\lambda_1 \leq \tilde{\lambda}_1 \leq \ldots \leq \lambda_n \leq \tilde{\lambda}_n$.

We also note that the trace and determinant of $C$ are commonly used measures of the overall variance and covariance for a given sample of data. Here, the model-parameter gradients $\nabla f_\theta$ constitute the data. Moreover, the trace and determinant of $\tilde{C}$ give a rough approximation of the same metrics of $C$. Proofs of these two properties are given in the supplementary material.

Thus, the summary statistics $\operatorname{tr} K_\theta$ and $\det K_\theta$ can be seen as approximations of the model gradient variance and covariance. *Smaller* values for the trace and determinant of the kernel, $K_\theta$, reflect a higher degree of model gradient similarity and thus a larger averaging effect in equation 3.

These metrics also serve as proxies for measuring the magnitude of the diagonal elements and relative magnitude of the diagonal-to-off-diagonal elements of $K_\theta$. From this perspective, smaller values for the trace and determinant of the $K_\theta$ reflect how well the kernel can be summarised by a low-rank representation. Low-rank $K_\theta$ can be viewed as indicative of coordinated learning in that it restricts the number of independent directions in which the functions can evolve in equation 3.

From a practical standpoint, the exact values of these metrics are not important. Rather, we wish to monitor how they evolve during training. Utilising the kernel also constitutes a pragmatic solution since calculating $C$, of size $p \times p$, is near infeasible for larger networks, whereas $K_\theta$ is only of size $n \times n$, where $p$ and $n$ are the number of parameters and data batch respectively, and generally $n \ll p$.

## 3    Tracking MGS Metrics During Training

Let us now investigate how the metrics $\operatorname{tr} K_\theta$ and $\det K_\theta$ evolve during training. Using these metrics, a wide range of regularisers, representing the most commonly used ones, are compared. Additionally, two new regularisation methods based on MGS are plotted alongside as a reference. These are introduced in the next section.

The results are shown in figures 2, 3, and 4. In each example, all methods have their hyperparameters tuned, using grid search and cross-validation. Additionally the results are aggregated after using different training splits and network initialisations. Figure 2 presents results from a simple fully-connected network (FCN) architecture trained on a toy problem generated from two concentric cirles perturbed with noise. Figure 3 stems from training a large convolution network (AlexNet) on a corrupted version of MNIST with label noise and restricted training size. Figure 4 is created in a similar way but instead uses the more difficult Fashion MNIST dataset [Xiao et al., 2017] and a network with residual connections (ResNet20) to verify if the phenomenon still occurs in more complex scenarios. In each example, despite being different problems and different architectures, all regularisation methods exhibit the same behaviour: when regularisation is applied the rate of MGS metric growth is decreased, indicating higher model gradient similarity. Furthermore, the test accuracy or generalisation performance (figures 3 & 4) is strongly correlated with the evolution of the MGS metrics. The slower the rate of MGS growth, the better the final generalisation, and when the growth plateaus so does the test accuracy.

We list some interesting observations from the MNIST example in figure 3:

- The unregularised network (gray) goes through a rapid boost in accuracy and then quickly overfits. When test accuracy peaks, there is a small plateau in the MGS metrics. However, once it starts to decline in accuracy, the MGS metrics again grow, indicating model gradient similarity is decreased.

- Weight penalty (blue) is able to regularise the network initially, but also eventually overfits. This is reflected in the MGS metrics as they plateau when test accuracy peaks and then start to increase at a seemingly proportional rate to the decline in performance.

- All methods that achieve high test accuracy control the growth of the MGS metrics. Still, many of them begin to overfit towards the end of training which coincides with the MGS metrics also gradually increasing.

- Only the MGS penalty (to be introduced in the next section) is able to maintain a stable test accuracy

- Dropout (brown) has gaps in its determinant metric. This means that it is causing $K_\theta$ to be singular which could have ramifications on the stability of the training.

More extensive test bench experiments are provided in section 5.

## 4    MGS regularisation

In the previous section, we saw a clear connection between boosting MGS (reflected by lower MGS metrics) and the ability of the network to generalise. The effect on MGS by the explicit regularisers was common to all the methods despite their widely differing design. Thus, we may think of each regularisation method as acting as a proxy for enforcing model gradient similarity. This motivates the construction of a new regularisation scheme that directly optimises MGS.

We thus modify the original loss to include a new penalty term $g(K_\theta(X)))$ which acts on the MGS kernel:

$$\widehat{\mathcal{L}}(f(X), Y) = \mathcal{L}(f(X), Y) + g(K_\theta(X))$$

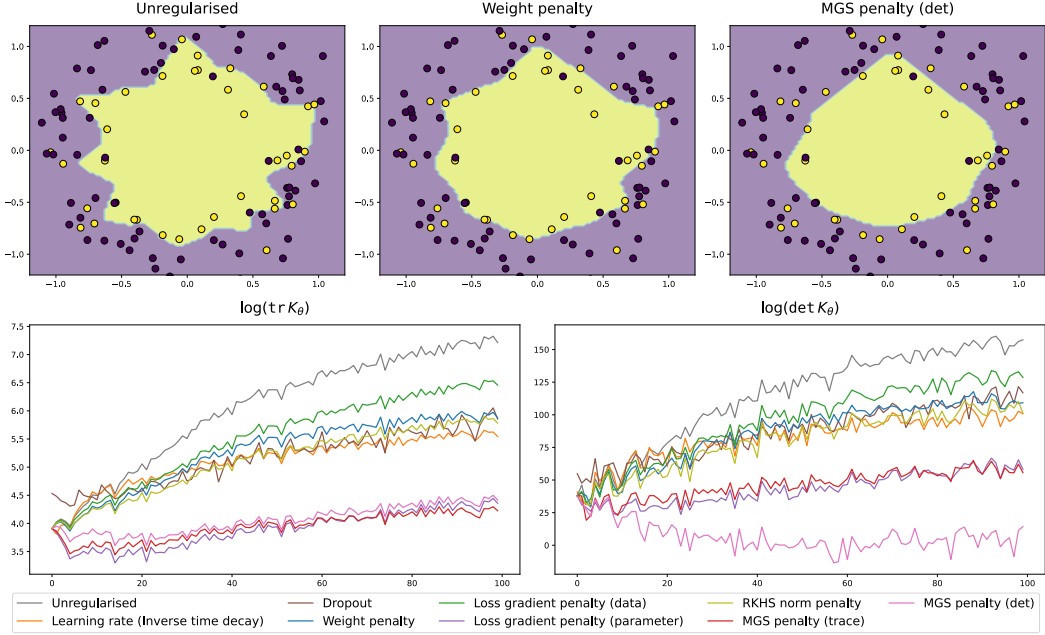

Figure 2: **FCN on two-circle classification. Top:** Decision boundaries are shown to visualise how well the network has generalised for each method. **Bottom:** MGS metrics vs. epochs for each regulariser are tracked during the training period. The connection between network generalisation and the MGS metrics is clear. For an unregularised network which overfits the data, MGS metrics grow rapidly. Once any explicit regularisation is used, MGS metric growth is slowed. Coupled with this, the decision boundaries approach the true model for methods that constrain the MGS metrics better. (Complete set of decision boundaries are provided in the supplementary material.)

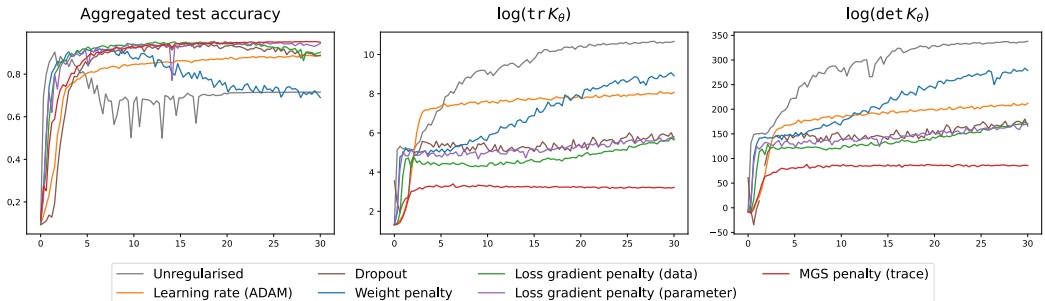

Figure 3: **AlexNet on corrupted MNIST. Left:** Test accuracy vs. training epochs. **Middle and Right:** MGS metrics vs. training epochs. Test accuracy is clearly coupled to MGS evolution. All methods that yield high final test accuracy exhibit small and slow-increasing MGS metrics, indicating a higher degree of model gradient similarity.

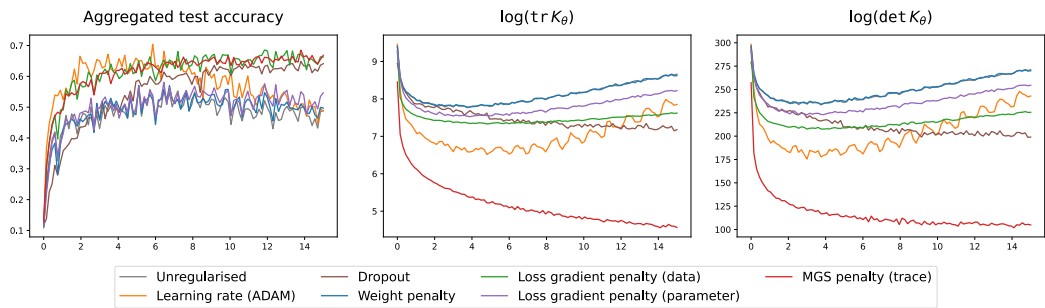

Figure 4: **ResNet20 on corrupted Fashion MNIST. Left:** Test accuracy vs. training epochs. **Middle and Right:** MGS metrics vs. training epochs. Even in this scenario using a more complex network, with residual connections, and a harder dataset, the coupling between test accuracy and MGS evolution is clear. Regularisers that achieve higher test accuracy also minimise the MGS metrics. Of note is the ADAM optimiser which first achieves good test accuracy but then quickly overfits, this is shown directly in the MGS metrics as well.

We already have two prime candidates for $g$, namely the previously investigated MGS metrics:

$$g_{\mathrm{tr}}(K_\theta(X)) = \alpha \operatorname{tr} K_\theta(X) = \alpha \sum_i \lambda_i$$

$$g_{\mathrm{det}}(K_\theta(X)) = \alpha \det K_\theta(X) = \alpha \log\left(\prod_i \lambda_i\right),$$

where $\alpha$ is a penalty factor and $\lambda_i$ are the eigenvalues from the spectral decomposition of $K_\theta$. As previously mentioned in section 2.3, smaller values for these penalty metrics reflect higher model gradient similarity. Adding an explicit penalty on these metrics thus forces the network to learn data in groups and preventing it from learning points individually (over-fitting or memorization).

The evolution of the two were shown in figures 2, 3 and 4. The metrics seem to behave similarly. However, $g_{\mathrm{tr}}(K_\theta)$ is more efficient to compute as it does not require either the complete spectral decomposition nor the full gradient similarity kernel to be computed. However, since both metrics are gradient based, they are, of course, computationally more burdensome than commonly used regularisers such as weight decay. For calculations of the penalties we make use of *Neural Tangents* [Novak et al., 2019] and *Fast Finite Width NTK* [Novak et al., 2022] libraries which are built on *JAX* [Bradbury et al. [2018]].

We note that the penalties are equivalent to penalising the arithmetic and geometric mean of the eigenvalues of the kernel, respectively. The former will mainly be affected by large leading eigenvalues. The latter essentially penalises the eigenvalues of the kernel on a log-scale and may thus take more of the overall structure of the kernel into account. Supplemental figure A.1 depicts results on the toy two-circle data from using both penalties. The differences are subtle. For pragmatic considerations, on real data we therefore utilise only the trace penalty for explicit regularisation, but both metrics can be used for monitoring neural network training.

Details on how $K_\theta$ is calculated with regards to mini-batches, vector-outputs and large data sets are provided in the supplementary material.

## 5   Experiments

We compare the performance of optimising MGS directly versus the most common, explicit regularisers. Although there exists a plethora of explicit regularisers and modifications thereof, the ones chosen are a representative of the main types of regularisers with the most widespread use in practice. As the aim is to compare performance between regularisers, a systematic investigation is done where the regularisers are rigorously tested against common overfitting scenarios caused by target noise and training size. Two main tests are done, one on a classification problem and the other on a regression problem. For each test two different types of architectures are chosen. This is to test how universal each method is in handling vastly different problems and networks. Finally, a test

Table 1: **Test accuracy for corrupted MNIST dataset using a LeNet-5 architecture.** Final test accuracy and one standard deviation is shown with the maximum test accuracy in parenthesis underneath. Noise column represents percentage of training labels that have been randomly flipped. Note the ability of MGS to handle large amounts of noise. Comparing max accuracy (in parenthesis) during training to final accuracy, MGS is also the most consistent and is not susceptible to overfitting.

| Noise | Unregularised | Dropout | Weight | Loss grad. | MGS |
|---|---|---|---|---|---|
| 0% | 89.6 ±1.3 (90.2) | **95.8** ±0.5 (95.9) | 83.1 ±9.6 (87.1) | 95.1 ±0.5 (95.1) | 95.2 ±0.3 (95.2) |
| 30% | 62.0 ±2.5 (85.2) | 80.9 ±2.9 (92.0) | 72.2 ±4.2 (79.2) | 88.7 ±2.0 (92.0) | **93.1** ±0.7 (93.4) |
| 60% | 37.3 ±2.9 (73.7) | 59.2 ±4.1 (83.7) | 10.0 ±0.4 (62.3) | 80.7 ±5.9 (81.4) | **88.4** ±1.4 (89.1) |
| 80% | 23.9 ±2.4 (62.4) | 39.5 ±4.7 (56.6) | 10.1 ±0.5 (23.5) | 15.1 ±5.4 (17.2) | **74.5** ±4.0 (75.1) |

of robustness to variability found in common training setups is performed. Complete details on the experiments, complete results and code to run them can be found in the supplementary material.

## 5.1 Classification: corrupted MNIST

We generate a corrupted version of the popular MNIST dataset by applying a motion blur to each training image. The test set is kept uncorrupted, which ascertains how well the network generalises: if the network overfits the training set, it will learn the corrupted version and perform poorly on the clean data. A testbench of challenging scenarios are created by varying the amount of label noise and training size. Each regularisation method is tuned in an identical setting, of medium difficulty, and then retains the chosen hyper-parameters for the entire testbench. This ensures an even playing field and that each method is given the same starting point. For each testing scenario, multiple runs are performed using a new network initialisation and resampling of the training data. No early stopping is used as we purely want to test how well each regularisation method can prevent over-fitting on its own.

Table 1 shows the results for 4 scenarios from the 143 scenario large testbench (see supplemental). In this scenario 3000 samples are used for training at 4 noise levels. It is immediately clear that MGS achieves the best performance overall in many aspects. Not only does it reach higher accuracy levels than the other methods, it is also robust in its performance. This can be seen both by the standard deviation of its final test accuracy, but also when comparing the max test accuracy to its final one. When comparing the maximum values, it is also evident that early stopping would not have resulted in better performance for any of the methods compared to the final MGS result. Additionally in supplemental figures A.2 and A.3, the evolution of test accuracy during training is depicted, visually showing that early stopping techniques do not exhibit improved performance over MGS. Indeed, MGS max and final test accuracy essentially coincide, indicating that MGS regularised networks do not overfit. The same conclusions hold true for a fully connected network (see supplemental) where performance is overall worse for all methods but MGS outperforms the others.

## 5.2 Regression: Facial Keypoints

A similar setup to the classification is created using corrupted version of the Facial Keypoints data set was used as a regression benchmark. Based on images of human faces, the problem is to predict the coordinates of 15 key facial features. Each image is first corrupted using a motion blur. Noise is then introduced by adding normally distributed values, using different scale parameters, to the target variables. Here, we fixed the number of training samples while the amount of target noise was varied. Each method is tuned using the same scenario first. Once again no early stopping is used. Similar conclusions can be drawn as from the classification results. The results are visible in table 2 where the noise column corresponds to the scale parameter of the normally distributed noise. Interestingly, only Dropout and MGS were actually able to converge to a satisfactory performance level. However,

Table 2: **Test loss for the corrupted Facial Keypoints dataset using a LeNet-5 architecture.** Final test loss and one standard deviation is shown with the minimum test loss in parenthesis underneath. Only Dropout and MGS achieve an acceptable level of final test loss. The other methods also provide a marginal increase in performance compared to the unregularised network. In the supplemental we compare results on a FCN architecture for which only MGS is able to attain good performance.

| Noise | Unregularised | Dropout | Weight | Loss grad. | MGS |
|---|---|---|---|---|---|
| 0 | 68.2 ±36.7 (54.2) | 14.9 ±2.2 (13.9) | 113.2 ±53.2 (90.3) | 212.4 ±174.1 (104.0) | **14.1** ±1.0 (13.8) |
| 10 | 54.8 ±24.0 (45.2) | 15.4 ±2.2 (14.4) | 92.6 ±46.5 (69.0) | 235.2 ±188.6 (140.5) | **14.3** ±1.1 (13.6) |
| 20 | 102.0 ±116.2 (48.2) | 16.6 ±2.2 (15.3) | 83.8 ±52.7 (49.3) | 266.1 ±233.1 (115.9) | **15.3** ±1.3 (14.7) |
| 30 | 94.8 ±131.5 (39.0) | 19.7 ±2.8 (17.7) | 70.6 ±58.0 (41.9) | 229.4 ±225.7 (151.5) | **17.8** ±2.0 (16.9) |

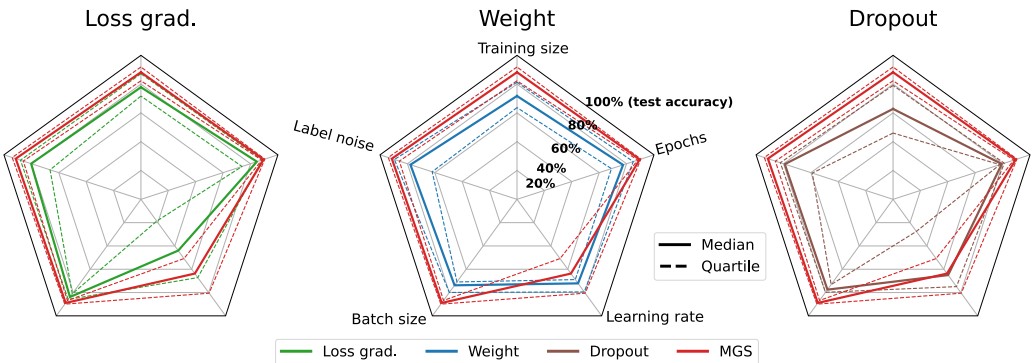

Figure 5: **Test accuracy quantiles after varying different parameters controlling training.** MGS is plotted in red against the other methods. MGS outperforms the other regularisers, and is also the most robust with regards to changes in the training setup, in all but one case. The only area in which it shows a degradation in performance is when the learning rate is changed. However this seems to affect all methods, but weight penalty to a lesser degree.

when results are compared for a FCN architecture (table A.2, supplemental), Dropout falls into the same category as the other methods while MGS performance remains high. This shows a weakness in a method such as Dropout: that it is inevitably architecture dependent.

## 5.3 Training parameter robustness

Finally, we test the robustness of each regularisation method by changing: amount of training size, label noise, batch size, learning rate, and epochs. A middle-ground scenario was chosen from this MNIST testbench as a starting point to tune each method. Then, the five training variables were changed individually to both higher and lower values, tracking the performance of each method after training. Consistent with the current findings, MGS outperforms each method substantially. In figure 5 we use radar charts to summarise the test bench results.

We note that MGS is the least affected by a change in all but one of the test parameters or conditions, and exhibits superior performance compared to the other regularisers. Performance is only affected by learning rate to some extent. This is not unexpected as the same can be seen for the other methods apart from weight penalty. Also, as is evident from the GD update step (equation 1), the one directly contributing training parameter in MGS is the learning rate.

# 6   Conclusion

We introduced the concept of *Model Gradient Similarity* (MGS) and discussed its connections to regularisation for models trained with gradient descent and the generalisation properties of neural networks. We proposed metrics that can be used to summarise MGS for a model and track the training of different neural network architectures for various learning problems.

It was shown that a wide range of explicit regularisers all appeared to attempt to enforce higher model gradient similarity, i.e. lower MGS metrics. Moreover, higher test accuracy performance was shown to be reflected in lower MGS metric values.

Based on these findings, a new type of regulariser, geared toward direct control of MGS, was designed and found to achieve top performance in several rigorous test bench experiments. Its overall robustness to label noise and training parameter settings was also an indication that directly optimising MGS comes closer to a more holistic approach to regularisation.

Taken together, these results provided insight into the underlying mechanisms of neural network regularisation. Due to the higher computational costs for gradient based regularisers, such as the MGS metric penalities introduced here or loss-gradient penalties, their use for direct optimisation is not always efficient. To scale for use in larger networks and in more complex settings, additional work is needed to obtain more efficient ways to compute or approximate the MGS kernel or metrics thereof. Future work could thus focus both on how MGS can be used to design new regularisers as well as to improve upon existing ones. The MGS metrics can be useful as KPI's for measuring a network's current capacity for under/over-fitting. Finally, the grouping effect of regularised neural network training, where model gradient similarity encourages coordinated learning across observations, suggests that MGS regularisation can be explored for joint prediction modeling and clustering.

## Acknowledgments and Disclosure of Funding

This research was supported by funding from Centiro Solutions, the Swedish Research Council (VR), the Swedish Foundation for Strategic Research, and the Wallenberg AI, Autonomous Systems and Software Program (WASP).

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
