# OpenReview forum: "On the Interpretability of Regularisation for Neural Networks Through Model Gradient Similarity"
_NeurIPS.cc/2022/Conference — NeurIPS 2022 Accept_

### Official Review · Reviewer_k97d · 2022-06-20

**Rating:** 6
**Confidence:** 3
**Soundness:** 3 good
**Presentation:** 2 fair
**Contribution:** 3 good

**Summary:**

This paper presents a novel view on regularization for neural networks. The core idea focuses on the similarity of model gradients when evaluating data points. The paper proposes a kernel that conveys the influence of one training sample has over the other. Two global metrics, the trace and the determinant, are used as proxy metrics to track the model generalization. They are also used as regularizers by including them in the optimization objective. Experimental results show that tracking and optimizing these metrics leads to models that retain a high generalization, even in the presence of noisy annotations for classification and regression problems.

**Questions:**

1. Figure 2: regularization methods are dependent on hyperparameters. Dropout depends on the probability of dropping out, weight decay is controlled by a small coefficient $\alpha_{\theta}$, etc. Are these results robust to the choice of these hyperparameters?
2. In general: the different regularizations depend on hyperparameters that are proper of each method. Those that are included in the loss (loss grad and weight decay) are modulated by a hyperparameter $\lambda$, similar to the way $\alpha$ is defined for $\widehat{\mathcal{L}}(f(X), Y)$ (equations between lines 175-177). Dropout depends on the probability of an activation being dropped (usually $p=0.5$). What protocol was followed to guarantee that those hyperparameters were optimal? This is particularly relevant for section 6.3 where the other hyperparameters were analyzed.
3. How dependent are these results on the batch size?
4. Corrupted MNIST: Section 6.1 mentions the use of motion blur, but table 1 only talks about flipped labels. What is considered a corruption in this case and what is the motivation behind adding motion blur?
5. Could MGS (i.e., $\text{tr}~K_{\theta}$) be used as an early stopping criterion?

**Limitations:**

There is little discussion of the limitations of this work, even though there are a few points that could be addressed:

- Even though the method is well supported by theoretical findings, a discussion on the effects of hyperparameters is lacking. In particular, effects of the batch size or assumptions about the data distribution e.g., the test set exhibits no covariance shift.
- From the empirical results, the model architecture lacks representative (and relevant) components that are often found in modern ANN architectures (see second point on “Significance”).
- A discussion of the experimental hyperparameters (how they were found and how they affect the results) is also missing in the main paper.
- Finally, an account on the computational costs of MGS metrics and the other baselines should be presented.

**Strengths And Weaknesses:**

- *Originality:*
    - Most of the elements in the paper, in isolation, are not new: the topic of regularization in ANNs has been studied for the past few decades. Tracking or including gradients (or other statistics) of data-batches to add constraints to the optimization process has also been done in the past. The idea of measuring the influence between samples during learning has also been addressed before. (In this regard, I was missing a discussion to the work on influence functions from Koh and Percy [1].) However, the originality from this work comes from the unique way to combine these ideas. In particular, the use of the determinant and the trace as global metrics.

- *Quality*:
    - In general, the main claims of the paper are well supported. A weak point can be found e.g., in the motivation (L39-40) where, the fact that one thing (regularization) can be achieved in multiple ways, doesn’t immediately implies that it is not well understood. One could argue that the more understood a problem gets, the more ways one could find to make it work.

- *Clarity*:
    - MGS was never defined nor tied to any metric or formula. (Saying that it is a “framework” is still vague, and needs a more concrete definition.) Something like “any measure of similarity that expresses the influence that one sample has on the model” will make MGS more tangible. Alternatively, MGS could be represented by the kernel $K_{\theta}$, while $\text{tr}$ and $\text{det}$ are the proposed metrics to track MGS.
    - Plots are small and most axes are not labeled.
    - Introducing MGS regularizers in section 5, while already presenting results based on those in section 4 is rather confusing. I strongly suggest restructuring the development of the paper so that it follows a more sequential timeline.
    - For the abstract, please use `\begin{abstract}...\end{abstract}`.
    - L143: does FCN stand for fully-connected network? Make sure it is explained somewhere to prevent confusions with “fully *convolutional* networks.”

- *Significance:*
    - Experiments using AlexNet (Figure 3): how representative is AlexNet w.r.t. state-of-the-art DL? Can architectural components such as residual connections, batch-normalization or initialization methods (He initialization vs. truncated gaussians) that are not part of AlexNet, affect the outcome of these experiments?
    - For a better overview of the viability of MGS metrics, it is important to report the computational cost (memory and FLOPs) of using MGS compared to dropout, weight-decay, etc.

## References

1. Koh, Pang Wei, and Percy Liang. "Understanding black-box predictions via influence functions." *International conference on machine learning*
. PMLR, 2017.

---

> ### Author Response · Authors · 2022-08-02
> **Author response to reviewer k97d**
>
> Thank you for detailed comments and thoughtful review! Below we answer all questions/concerns that you had:
>
> ### Originality
> We have added small discussion on the use of influence functions by Koh and Percy.
>
> ### Quality
> > ...the fact that one thing (regularization) can be achieved in multiple ways, doesn’t immediately implies that it is not well understood...
>
> This is a very good point, and we have now removed that statement.
>
> ### Clarity
> > MGS was never defined nor tied to any metric or formula...
>
> We refer to MGS as a "framework" or concept due to it having many use cases, e.g. monitoring metric, explicit regulariser, etc. Therefore, we found it more intuitive to call it a framework as that is in line with the goal of this paper which is to present MGS as a new way of understanding regularisation and a vehicle to propose new methods. The order of sections 2 and 3 is perhaps a little unconventional, but was chosen deliberately so that the emphasis of the paper doesn't solely become the two explicit MGS regularisers we investigated here.
>
> > Plots are small and most axes are not labeled.
>
> We felt that the plots would become even more cramped with explicit labels. To make it clearer though, we have added some extra explanations to all captions on exactly what is being plotted against what. The plots being small is a valid concern however making them bigger would require moving them onto separate lines which in turn makes it difficult to compare the different plots side-by-side, which is important to see the correlation between test performance and MGS metrics.
>
> > Introducing MGS regularizers in section 4 while already presenting results based on those in section 3 is rather confusing...
>
> Agreed that this can be slightly confusing (see also above). However, we found no efficient way of first presenting the observations about the MGS metrics, then the MGS regularisers, and finally comparing them back against the initial observations without duplicating plots.
>
> > For the abstract, please use `\begin{abstract}...\end{abstract}`
>
> Thank you for noticing. We have now fixed this!
>
> > L143: does FCN stand for fully-connected network?
>
> This has been fixed and the complete definition (fully connected network) has been added before the first use of the FCN abbreviation.
>
> ### Significance
> > Experiments using AlexNet (Figure 3): how representative is AlexNet w.r.t. state-of-the-art DL...
>
> In general we don't expect the architecture to influence the main findings about MGS. The core behind MGS in equations 1, 2, and 3 does not specify any particular type of model. For completeness, however, we have added an extra experiment using a more complex network with residual connections (ResNet20) trained on a harder dataset (Fashion MNIST) which shows that the current findings about MGS still hold even in more complicated scenarios (new figure 4). Also another minor note, specifically for the dropout results in the testbenches, batchnorm was used between convolutional layers for LeNet-5.
>
> > For a better overview of the viability of MGS metrics, it is important to report the computational cost...
>
> The two MGS regularisers proposed in this paper fall into the same class as other "double back-propagation" methods in terms of memory and computational costs. Therefore we did not feel that listing explicit computational costs to be of high interest. Generally this class of methods is not used as-is and instead optimised versions of them are derived instead. We expect this to also be the case for MGS based regularisers, and look forward to designing performant ones that operate on the same principle of directly increasing MGS in future work.
>
> ### Questions
> > 1 & 2: regarding hyperparameters
>
> In all experiments, including the test-benches, each method goes through the same hyper-parameter tuning procedure which entails using grid-search with cross-validation. To ensure that this is clear to the reader, we have expanded upon this discussion in the paper.
>
> > 3: How dependent are these results on the batch size?
>
> In section 5.3 we show that MGS is the least affected when changing batch size (see figure 5).
>
> > 4: What is considered a corruption in this case and what is the motivation behind adding motion blur?
>
> We have added some new text to make it clear that it is only the training set that is corrupted and why that is done. The reason for doing this is to properly test each regularisation method as the unregularised network will most likely learn features for the corrupted data which are not necessarily suited for predicting on the clean data. Regularisation should help the network to generalise and therefore learn features from the corrupted data that still also work well on the clean dataset.
>
> > 5: Could MGS be used as an early stopping criterion?
>
> A good observation, and definitely something we think could be done, however we have left it for future work.
>
> ### Limitations
>
> All concerns regarding limitations have been answered above.

---

### Official Review · Reviewer_TsWy · 2022-06-25

**Rating:** 3
**Confidence:** 4
**Soundness:** 3 good
**Presentation:** 4 excellent
**Contribution:** 1 poor

**Summary:**

the submission proposed to use the trace of the kernel matrix, which is constructed by taking the inner product of a matrix that has gradients of function outputs w.r.t. parameters and the transpose of the matrix itself, to regularise the training of a neural network. The submission through experiments showed that their proposed gradient-based regulariser provided better generalisation than other existing regularisation approaches.

**Questions:**

See above.

**Limitations:**

See above.

**Strengths And Weaknesses:**

1. the title of the submission doesn't seem to reflect the content of the submission.

The analysis and the proposed regulariser eventually don't convey much information on the similarity of gradients computed on a pair of data samples, since it is purely based on the l2 norm of gradient vectors on individual data samples. Even though the determinant of a matrix utilises the pair-wise similarity, as the submission mentioned as well, the difference between the effect of trace and that of the determinant is subtle.

2. I am not sure how the gradient is used when there are multiple outputs, for example, multi-class classification problems.

For classification problems with more than two output classes, the gradient of the function w.r.t. the parameter on each data sample is a matrix. It doesn't seem clear to me how the proposed approach uses the matrix.

3. The proposed regulariser is similar to the Lipschitz constraint that has been used in Wasserstein GANs, and other regularisation schemes.

The idea of Lipschitz constraint is to make sure that a small change in the input data sample only leads to a tiny change in the output, and the proposed approach in this paper, which computes the trace of the kernel matrix, regularises / penalises the norm of a gradient vector, which is essentially regularises the Lipschitz constant of the model locally.

A relatively thorough theoretical comparison can be found here: https://proceedings.neurips.cc/paper/2020/file/8929c70f8d710e412d38da624b21c3c8-Paper.pdf

4. In all experiments, I wonder if any early stopping is applied, and if not, I think early stopping could alleviate demonstrated overfitting issues.

It is a little wild to me that weight regularisation and dropout performed very poorly, but I think realistically, early stopping is well-adopted on top of various regularisation schemes, and it could reduce the performance between the proposed approach and all other methods.

---

> ### Author Response · Authors · 2022-08-02
> **Author response to reviewer TsWy**
>
> Thank you for your questions. We have taken your objections into consideration and added clarifying text in places to ensure that readers do not miss that e.g. multi-class cases and early stopping are indeed discussed and investigated in the paper. However, with respect to your first question, the link between MGS and GD is already discussed at length in section 2.
>
> > the title of the submission doesn't seem to reflect the content of the submission.
>
> As written in section 2.2 and 2.3, MGS has an additional contribution to GD itself. If the trace (diagonal elements) is large then the averaging effect caused by the pairwise elements (off-diagonal elements) will be negligible and the model will learn each data point independently. Penalising the trace, diminishes the diagonal, and therefore increases the strength of the averaging effect from the pairwise elements. This, in turn, encourages the network to find relevant groupings in the data, based on gradient similarity, that still minimise the empirical loss. On the other hand, penalising the determinant results instead in the pairwise elements to grow, which directly increases the averaging effect as well. Thus both the trace and determinant contribute to increasing gradient similarity but in slightly different ways.
>
> > I am not sure how the gradient is used when there are multiple outputs, for example, multi-class classification problems.
>
> We state on L202-203 (close of section 4) that details on how $K_\theta$ is calculated with regards to mini-batches, **vector-outputs**, and large data-sets are included in the supplementary material. In general we use the same methods found in corresponding Neural Tangent literature [1] (and software libraries) where the gradients for multiple outputs are concatenated prior to calculating the inner product.
>
> > The proposed regulariser is similar to the Lipschitz constraint that has been used in Wasserstein GANs, and other regularisation schemes.
>
> The gradients are calculated with regards to the model parameters and not the input data (as already identified in your summary). This is not the same as the gradient penalty, used for example in WGANS to implement a Lipschitz constraint, which calculates the gradient w.r.t. input data [2]. Nor do we use MGS to explicitly improve data distribution (perturbations in input data) robustness, though robustness to data corruption is implicitly achieved by the grouping effect of the MGS regularisers.
>
> > In all experiments, I wonder if any early stopping is applied, and if not, I think early stopping could alleviate demonstrated overfitting issues.
>
> Early stopping is investigated and results are provided in the supplementary materials. There, we already include plots that show the evolution of the test accuracy during training for a selection of the test-bench runs. It is clearly visible that early stopping would not achieve the best performance, and in many instances is significantly worse than what regularisation with MGS achieves. In the main manuscript experiments we don't use early stopping in order to illustrate that MGS is the only method that effectively combats over-fitting while still achieving good performance. These results are shown in plots in figures 2-4, A2, and A3. Note also that maximum test accuracies (minimum loss) during training are provided in Tables 1 and 2.
>
> > It is a little wild to me that weight regularisation and dropout performed very poorly, but I think realistically, early stopping is well-adopted on top of various regularisation schemes, and it could reduce the performance between the proposed approach and all other methods.
>
> Firstly, we disagree that dropout and weight decay perform very poorly. In our testbench results, dropout is usually next best and weight decay works quite well at low- to moderate noise levels. As mentioned above, we did investigate early stopping and max accuracies over training are reported in Tables 1 and 2. Early stopping did marginally improve performance for both dropout and weight decay for several cases. However in Tables 1 and 2 we especially note that MGS is the only method for which final and max accuracies essentially coincide, and for most cases at a higher level than max accuracy for the other regularisers.
>
> 1. Jaehoon Lee, Lechao Xiao, Samuel Schoenholz, Yasaman Bahri, Roman No-vak, Jascha Sohl-Dickstein, and Jeffrey Pennington. Wide neural networks of any depth evolve as linear models under gradient descent. Advances in Neural Information Processing Systems, volume 32, pages 8572–8583.
>
> 2. Ishaan Gulrajani, Faruk Ahmed, Martin Arjovsky, Vincent Dumoulin, and Aaron Courville. Improved training of wasserstein gans. In Proceedings of the 31st International Conference on Neural Information Processing Systems, NIPS’17, page 5769–5779.

---

### Official Review · Reviewer_1j14 · 2022-07-11

**Rating:** 7
**Confidence:** 3
**Soundness:** 4 excellent
**Presentation:** 4 excellent
**Contribution:** 4 excellent

**Summary:**

In this paper, the authors proposed a new concept called "Model Gradient Similarity" (MGS), which can be used to analyze the gradient descent algorithm and the effect of explicit regularizations, and can be used to design new regularization schemes. The main contributions of this paper are as follows:

(1) The authors defined MGS as the dot product between two model-parameter gradient vectors from two different training inputs -- this can be viewed as a kernel. The authors did an analysis based on first-order Taylor decomposition, and derived a first-order approximation of how the model behavior would change after one update of gradient descent. In essence, the (approximate) amount of change is (the negative of) the learning rate, multiplied by the matrix vector product between the MGS kernel matrix and the loss-model gradient vector.

(2) The authors further developed two metrics to summarize the MGS, by using the trace and the determinant of the MGS kernel matrix computed from training data. These can be used to approximate variance and covariance of model gradients on training data, and provide summary statistics of MGS -- the smaller these metrics, the larger the MGS is.

(3) The authors tracked the MGS metrics during training for several regularization schemes, and discovered that all of them to some extent function by trying to inhibit the growth of MGS metrics (i.e., encouraging higher MGS among training samples).

(4) Finally, the authors designed explicit regularization schemes, one based on the trace and the other the determinant of the MGS kernel matrix on training data and showed that (1) both worked well on a toy example, and (2) the trace regularization can be applied to real-world datasets to improve both accuracy and robustness -- for computational reasons only trace is applied to real datasets.

**Questions:**

I do not have questions -- the paper is very well written and everything is explained clearly.

**Limitations:**

The authors have adequately addressed the limitations of their work, namely, the computationally intensive nature of gradient-based regularization schemes, and also in particular, of the proposed MGS determinant regularization.

**Strengths And Weaknesses:**

Strengths:

+ The first-order Taylor decomposition analysis on gradient descent to show the effect of MGS on model outputs is very neat. Given that the learning rate is usually small in practice, the change in parameters after one update is generally small in norm, so the first-order analysis generally holds. Great work!

+ The summary metrics to quantify MGS are theoretically grounded and sound.

+ The observation that all explicit regularization schemes inhibit MGS metrics (trace and determinant) growth is also insightful, and sheds light on why regularization works.

+ The proposed MGS trace regularization is simple and effective.

Weaknesses:

There are not many weaknesses. If possible, I would love to see a mathematical analysis of how MGS is linked to regularization schemes such as weight decay -- but this can be future work.

There is one small typo that needs to be fixed:

Equation (2) contains an error after the "approximately equal" sign. It should be f_\theta(x'), rather than f_\theta(x).

---

> ### Author Response · Authors · 2022-08-02
> **Author response to reviewer 1j14**
>
> Thank you for the thorough and well-structured review! Below we answer the questions/concerns you had:
> > I would love to see a mathematical analysis of how MGS is linked to regularisation schemes such as weight decay.
>
> This is exactly what we would like to do in future work. To be able to link existing regularisation schemes to MGS analytically would help in deriving more efficient methods for example.
>
> > Equation (2) contains an error.
>
> Thanks for spotting this, we have fixed it in the rebuttal revision.

---

> > ### Comment · Reviewer_1j14 · 2022-08-09
> > **Thank you for the response**
> >
> > Thank you for the response and fixing equation (2).

---

### Meta-Review · Area_Chair_6rdu · 2022-08-28

**Recommendation:** Accept
**Confidence:** Less certain

**Metareview:**

This paper is controversial among the reviewers. On the positive side, reviewers like the novelty of the concept, the derivations and the clear presentation. The negative review wonders why the proposed method performs much better than dropout, similarity to Lipschitz constraints, and whether proper early stopping was used. The authors addressed some of the concerns, though the reviewer was not convinced. In the AC's opinion, the objections are sufficiently addressed and are not clear enough for rejection. One reviewer wanted to know more about the computational costs of using this technique and more discussions on the limitations.

The paper proposed a novel and interesting approach to regularization and seems to be a good contribution to the community.

**Award:**

No

---

### Decision · Program_Chairs · 2022-09-14

Accept